# Systemic Inflammation and Oxidative Stress in Childhood Obesity: Sex Differences in Adiposity Indices and Cardiovascular Risk

**DOI:** 10.3390/biomedicines13010058

**Published:** 2024-12-29

**Authors:** Tjaša Hertiš Petek, Evgenija Homšak, Mateja Svetej, Nataša Marčun Varda

**Affiliations:** 1Department of Pediatrics, University Medical Centre Maribor, Ljubljanska ulica 5, 2000 Maribor, Slovenia; tjasa.hertispetek@ukc-mb.si; 2Department of Laboratory Diagnostics, University Medical Centre Maribor, Ljubljanska ulica 5, 2000 Maribor, Slovenia; evgenija.homsak@ukc-mb.si (E.H.); mateja.svetej@ukc-mb.si (M.S.); 3Faculty of Medicine, University of Maribor, Taborska ulica 8, 2000 Maribor, Slovenia

**Keywords:** childhood obesity, cardiovascular risk, inflammation biomarkers, oxidative stress, adiposity indices, cardiometabolic profile, sex differences, myeloperoxidase, I-TAC/CXCL11

## Abstract

**Background:** Systemic inflammation and oxidative stress are fundamental contributors to the onset of conditions related to childhood obesity, such as cardiovascular (CV) diseases. We aimed to assess CV risk in childhood obesity by examining sex differences in adiposity indices, cardiometabolic profiles, inflammation, and oxidative stress biomarkers. We also aimed to assess the potential of the interferon-inducible T-cell alpha chemoattractant (I-TAC/CXCL11) as a novel biomarker. **Methods:** Eighty children (36 girls) aged 5–18 years with overweight, obesity, or normal weight were analyzed. Fasting blood samples were obtained to assess C-reactive protein (CRP), leukocytes, myeloperoxidase (MPO), adiponectin, monocyte chemoattractant protein-1, superoxide dismutase-1, I-TAC/CXCL11, and a comprehensive cardiometabolic profile, including glucose, lipid, renal, liver, and thyroid function markers. Adiposity indices were determined using bioelectrical impedance analysis (BIA) and anthropometric measures, including BMI, waist-to-hip and waist-to-height ratios, and visceral and subcutaneous fat thickness. Blood pressure (BP) and pulse wave velocity were also evaluated. **Results:** Girls had less central obesity and fewer CV risk factors than boys, despite having similar total fat mass. Both girls and boys with overweight or obesity showed higher CRP levels. Girls with excess weight had increased leukocyte counts, while boys had elevated MPO levels, which correlated positively with adiposity indices, systolic BP, and homocysteine, and negatively with HDL. I-TAC/CXCL11 levels were similar across groups. **Conclusions:** Adiposity indices are essential for evaluating CV risk in children and adolescents, with sex differences underscoring the need for tailored approaches. MPO correlated significantly with CV risk markers, supporting its inclusion in routine assessments. I-TAC/CXCL11 warrants further study in childhood obesity.

## 1. Introduction

Obesity has risen to epidemic proportions in both adults and children, contributing to the onset of cardiovascular (CV) diseases and various other non-communicable diseases. While certain obesity-related complications are less common in children and adolescents than in adults, they are likely to substantially raise morbidity and mortality across the entire population over time [1,2,3]. Interestingly, despite the observed sex differences, relatively few studies have specifically explored variations in childhood obesity [4].

Body mass index (BMI) is a key metric for evaluating obesity, which is a significant risk factor for CVD [5]. Although BMI is widely used to assess excess weight, it does not account for body composition (percentage of body fat) or the distribution of fat in the body [6]. Several methods are available for assessing body composition, including anthropometry, dual-energy X-ray absorptiometry, and bioelectrical impedance analysis (BIA) [7]. BIA is recognized as an effective, early, and non-invasive bedside technique for evaluating CV risk in children [8]. However, its limitation is that it cannot assess fat distribution [9].

Independent of BMI or overall body fat percentage, the amount of abdominal fat—especially visceral fat—and liver fat are most closely linked to fat tissue inflammation and metabolic disorders [9]. Ultrasound has proven valuable for distinguishing and measuring visceral and subcutaneous fat and is regarded as a useful tool for assessing abdominal obesity in children and adolescents [10]. Waist circumference (WC) and waist-to-hip ratio (WHR) are also reliable indicators of abdominal obesity and are strongly associated with health complications [6]. Furthermore, measuring WC is important for identifying metabolic syndrome, which is characterized by central adiposity, dyslipidemia, arterial hypertension, and impaired glucose regulation, all of which are strong indicators of increased CV risk [11].

Childhood obesity and overweight have been positively associated with chronic systemic inflammation and oxidative stress [12,13], both of which appear to be key factors in the development of obesity-related disorders in children, such as early atherosclerosis [11]. Common sites of atherosclerosis in children are the abdominal aorta, carotid, coronary, and renal arteries [14]. Atherosclerosis is closely associated with arterial stiffness [15], which can be assessed through pulse wave velocity (PWV)—an established method for evaluating CV risk in children and adolescents [16].

Laboratory tests, including lipid profiles, fasting glucose, and liver enzyme levels, are critical for detecting early signs of metabolic syndrome and non-alcoholic fatty liver disease, both increasingly linked to childhood obesity [17,18]. Additionally, markers such as apolipoproteins A1 and B, homocysteine, urate, and renal function indicators like cystatin C may provide further insight into elevated CV risk [19].

Although tests for inflammation and oxidative stress biomarkers are not yet standard in clinical practice, they hold promise for enhancing risk assessment and improving patient care. Biomarkers that are commonly used in research to assess inflammation and oxidative stress in relation to CV risk include adiponectin (ADPN), monocyte chemoattractant protein-1 (MCP-1), myeloperoxidase (MPO), and superoxide dismutase-1 (SOD-1), among others [11]. Furthermore, findings also suggest that routine laboratory parameters, such as total leukocyte/white blood cell (WBC) count and C-reactive protein (CRP), may serve as reliable markers of inflammation in children and adolescents with obesity [20,21,22,23]. Additionally, a newer biomarker of inflammation in obesity is Interferon-inducible T-cell alpha chemoattractant (I-TAC/CXCL11), a member of the CXC chemokine family involved in recruiting T lymphocytes and other immune system cells to inflammation sites. In adult population, I-TAC has been linked to endothelial white blood cell stasis, heightened CV risk in obesity, and adipose tissue angiogenesis [24,25]. However, studies on I-TAC in obesity among adults remain limited, and its relevance to pediatric obesity is still not well explored, as discussed in our prior study.

Our study aimed to investigate both conventional biomarkers of inflammation and oxidative stress, as well as the novel inflammatory marker I-TAC, with respect to sex-related differences in obesity indices such as body composition, fat distribution, and overall CV risk assessment in the pediatric population with overweight and obesity.

## 2. Materials and Methods

### 2.1. Study Description

We conducted a cross-sectional study in a cohort of children and adolescents to investigate the associations between inflammation, oxidative stress parameters, laboratory tests for cardiovascular risk assessment, body composition, fat distribution, blood pressure, and arterial stiffness measurements (PWV). The aim was to enhance sex-related cardiovascular risk assessment and explore the potential role of the selected biomarkers in the pediatric population. The study adhered to the ethical principles outlines in the Helsinki Declaration. The research protocol and procedures received approval from both the institutional ethics board (The Medical Ethics Committee of the University Medical Centre Maribor) and the national ethics board (The National Medical Ethics Committee of the Republic of Slovenia).

### 2.2. Sample

The sample size was determined using an online calculator available on the Cleveland Clinic website, based on a published article [26], with an alpha value of 0.05 and a power of 80%. A total of 82 children, aged 5 to 18, were enrolled based on their outpatient visits at the University Medical Centre Maribor, Slovenia, Department of Pediatrics. Participants were assessed for potential exclusion criteria, such as ongoing therapy, dietary supplements, acute infection and other comorbidities that could affect cardiometabolic health or inflammation/oxidative stress indices. Two participants were excluded due to the need for regular treatment, which was not disclosed during the initial screening. The study population was described in more detail in a previously published article investigating the role of vitamin D [27]. Informed written consent was obtained from the parents or legal guardians of participants, or directly from those aged 15 and older. All participants were informed of the voluntary nature of the study and were assured that their data would remain confidential.

### 2.3. Data Collection

Data were collected from October 2022 to May 2023. Fasting venous blood samples were obtained to assess the following parameters: urea, creatinine, cystatin C, fasting glucose (FG), total cholesterol (TC), triglycerides (TGC), high-density lipoprotein (HDL), low-density lipoprotein (LDL), apolipoproteins A1 (ApoA1) and B (ApoB), lipoprotein (a) (Lp(a)), homocysteine (Hcy), and urate. Liver function tests were also performed, including aspartate aminotransferase (AST), alanine aminotransferase (ALT), gamma-glutamyl transferase (gamma-GT), and total bilirubin (TB). Thyroid profile tests (thyroid-stimulating hormone [TSH], triiodothyronine [T3], and thyroxine [T4]) were measured. C-reactive protein (CRP) and leukocytes (WBC) levels were assessed, along with selected markers of chronic systemic inflammation and oxidative stress: adiponectin (ADPN), monocyte chemoattractant protein-1 (MCP-1), myeloperoxidase (MPO), superoxide dismutase-1 (SOD-1), and interferon-inducible T-cell alpha chemoattractant (I-TAC/CXCL11). All selected biomarkers of chronic systemic inflammation and oxidative stress were measured using enzyme-linked immunosorbent assay (ELISA) kits. For the determination of ADPN, MCP-1, MPO, and I-TAC, ELISA kits from R&D Systems (Minneapolis, USA) were used. The SOD-1 levels were measured using an ELISA kit provided by CUSABIO (Houston, USA).

Body mass (kg), body height (cm), and waist and hip circumferences (cm) were measured in accordance with World Health Organization guidelines [28]. WC was taken at the narrowest point of the waist between the lower ribs and the navel, while HC was measured at the widest part of the children’s hips. A tape measure with 0.1 cm precision was used for circumferences. Height was recorded using a stadiometer attached to the scale, and body mass was determined with a digital scale accurate to 0.1 kg. Waist-to-hip ratio (WHR) and waist-to-height ratio (WHtR) were calculated by dividing waist circumference by hip circumference and by dividing waist circumference by height, respectively.

BMI was calculated as weight in kilograms divided by height in meters squared (kg/m^2^) and the corresponding percentiles were derived based on the growth charts provided by the Centers for Disease Control and Prevention (CDC) [29]. Children with BMI ≥5th and <85th percentile for their age and sex were classified as having normal weight, those ≥85th percentile as overweight, those ≥95th percentile as obese, and those ≥120% of the 95th percentile as having morbid/severe obesity [29,30].

Body composition was assessed using bioelectrical impedance analysis (BIA) with the Nutrilab Bioimpedance device Akern 2016 and Biatrodes Akern electrodes. Measurements were performed under standardized conditions: participants were in a fasting state, had emptied their bladder, and were lying supine. To ensure consistency, proper electrode placement was observed, and participants refrained from wearing jewelry near the electrodes, engaging in exercise within 12 h, or consuming caffeine or alcohol intake within 24 h. Parameters analyzed with the manufacturer’s software (Bodygram Plus 1.2.2.8, Akern, Florence, Italy) included phase angle (PA in °), fat mass (FM in kg), body cell mass (BCM in kg), fat-free mass (FFM in kg), total body water (TBW in L), and extracellular water (ECW in L). PA represents the ratio of resistance (R) to reactance (Xc). FM measures adipose tissue storage containing glycerol and fatty acids. BCM quantifies metabolically active, protein-based tissue. FFM is derived by subtracting FM from total body weight. TBW reflects both intracellular and extracellular water, influenced by age, sex, and muscle mass. ECW refers to fluid outside cells, including that in vessels and lymphatics. The procedure followed recommendations from the Akern Bodygram Plus Software Guide (Version 1.2.2.8, Akern, Florence, Italy) [8].

Abdominal ultrasounds were conducted by trained physicians using an abdominal transducer to assess visceral and subcutaneous fat thickness. The assessment followed the standardized protocol for evaluating abdominal fat in both pediatric and adult populations [31,32]. Visceral fat thickness was defined as the distance between the peritoneum and the anterior surface of the lumbar vertebral body, while subcutaneous fat thickness was measured as the distance from the skin to the ventral margin of the abdominal muscles (linea alba).

Systemic blood pressure was recorded once using a digital automatic device (Omron^®^; Osaka, Japan) while the subject was seated, after 5 min of rest. Blood pressure values for boys and girls were interpreted based on age and height percentiles [33]. Pulse wave velocity (PWV) was assessed using arterial applanation tonometry (SphygmoCor, SCOR-Vx, Sydney, NSW, Australia). Trained investigators conducted all measurements to ensure high intraobserver reliability. Prior to measurements, the participant’s height, weight, blood pressure, and the arterial path length between the radial and carotid arteries) were recorded. A pressure tonometer was then used to transcutaneously capture the pressure pulse waveform, with an ECG signal providing a timing reference for software-based PWV calculation [34].

### 2.4. Statistical Analysis

Statistical analyses were performed using IBM SPSS Statistics (Version 29), R (Version 4.3), and Jamovi (Version 2.5). Descriptive statistics were used to summarize demographic characteristics. The normality of the data was assessed using the Shapiro–Wilk test. Since some variables were not normally distributed, the Mann–Whitney test was used for comparisons. The sample was first stratified by sex. Further stratification was performed based on BMI percentile (pBMI), with normal weight defined as the 5th percentile to less than the 85th percentile, overweight as the 85th to less than the 95th percentile, obesity as greater than the 95th percentile, and morbid/severe obesity as ≥120% of the 95th percentile, corresponding to adult obesity types 2 and 3. Comparisons within each subgroup were also conducted using the Mann–Whitney U test. Results were presented as median (IQR). A *p*-value < 0.05 was considered statistically significant. Statistically significant differences between groups were denoted by *. Spearman’s rho correlation analysis was conducted to explore associations between inflammation/oxidative stress parameters, adiposity indices, laboratory parameters, blood pressure and pulse wave velocity. Meaningful relationships were defined as those with a correlation coefficient greater than 0.3 and *p*-values < 0.05. All *p*-values were adjusted using the Benjamini–Hochberg (BH) correction to account for multiple comparisons.

## 3. Results

### Sample Characteristics

A total of 80 children and adolescents, aged 5 to 18 years, participated in the study, comprising 36 girls (45%) and 44 boys (55%). Adiposity indices, blood pressure, pulse wave velocity, laboratory parameters for cardiovascular risk assessment, and biomarkers of inflammation and oxidative stress were analyzed by group (girls vs. boys).

Girls, on average, had lower weight, height, and WHR compared to boys. They also exhibited better overall fat distribution, with lower US-measured visceral fat thickness and waist circumference. Additionally, girls had lower levels of FFM, TBW, ECW, and BCM. Interestingly, there were no significant differences in fat mass (FM) between girls and boys (Table 1).

Boys and girls exhibited similar BP levels and PWV values. When stratified by blood pressure percentiles categories, the findings were consistent between sexes as well (Appendix A). Laboratory tests showed significant differences, with girls having lower liver enzymes (AST and ALT), lower uric acid, lower urea, lower cystatin C, lower TSH and T3, and lower Hb levels. Overall, girls accumulated fewer cardiovascular risk factors compared to boys (Table 2).

Inflammation and oxidative stress biomarkers showed no significant differences between boys and girls when analyzed without stratification by pBMI (BMI percentile) (Table 3).

When the sample was stratified based on sex and pBMI, thirty children (13 girls) were in the normal weight group, with BMI values between the 5th and 85th percentiles. Five children (four girls) were classified as overweight (BMI above the 85th percentile), while 45 children (19 girls) had BMI values above the 95th percentile and were categorized as obese, of which 22 (six girls) were classified as having morbid/severe obesity (pBMI ≥ 120% of the 95th percentile or greater).

Both boys and girls with overweight or obesity had significantly higher CRP levels. Girls with excess pBMI also exhibited increased leukocytes compared to normal-weight girls, a difference not observed in boys. Interestingly, boys with overweight or obesity had higher levels of MPO. Levels of adiponectin, MCP-1, and I-TAC were similar across all groups. Values for SOD-1 were not reported as they were below the detection limit (Table 4). The results are graphically presented in Figure 1 for girls and Figure 2 for boys.

Spearman’s rho correlations between inflammation and oxidative stress markers, adiposity indices, laboratory parameters for cardiovascular risk evaluation, blood pressure, and pulse wave velocity are illustrated in Figure 3.

MPO positively correlated with adiposity indices such as anthropometric measurements—weight, pBMI, BMI, WC, HC, WHtR, VFT, SFT and body composition parameters—FM, FFM, BCM, and TBW. There was also a positive correlation with homocysteine and a negative correlation with HDL. Importantly, MPO was positively correlated with SBP and showed a positive trend toward increased PWV (ρ = 0.30, *p* = 0.053).

ADPN exhibited a positive correlation with HDL and a negative correlation with BMI and WC. I-TAC displayed a negative correlation with SBP. WBC showed a positive correlation with WHtR, while CRP correlated positively with WHtR, BMI, pBMI and WC. MCP-1 showed no significant correlations.

The overall state of inflammation and oxidative stress (as indicated by ADPN, MCP-1, MPO, I-TAC, SOD-1, CRP, and WBC) was assessed using principal component analysis (PCA) [27]. Inflammation and oxidative stress showed a positive correlation with weight, BMI, pBMI, WHtR, WC, HC, UZ-measured VFT and SFT at abdomen, bioimpedance-measured FM, and negative correlation with HDL. Interestingly, there was also a positive relationship with PWV, indicating stiffening of arteries in relation to increased oxidative stress and inflammation.

## 4. Discussion

Childhood obesity has become a pressing public health concern worldwide, significantly impacting CV health [11]. Obesity is usually accompanied by inflammation of fat tissue, with a prominent role of visceral fat [32]. Recent research underscores the expanding recognition of the importance of anthropometric measurements, body composition, and fat distribution markers in understanding the development of inflammation/oxidative stress and their role in CV risk assessment [11,35]. Our aim was to investigate these risk factors in relation to both traditional biomarkers and a novel biomarker of inflammation and oxidative stress, with the objective of improving sex-adjusted CV risk assessment in the pediatric population with obesity.

Excess body fat (adiposity), particularly in the abdominal region, poses significant health risks. While BMI remains the most widely used measure of excess weight, alternative indicators like WC, WHR, and WHtR provide superior insight into fat distribution and central obesity, with stronger links to morbidity and CV risk [6]. Notably, abdominal obesity, as defined by WC, has been identified as an independent CV disease risk factor, distinct from BMI [36]. Because BMI does not differentiate between fat and muscle mass, recent research supports a broader approach, integrating tools such as abdominal circumference and visceral fat assessments to better classify diverse obesity phenotypes [36,37]. On average, girls in our cohort exhibited better fat distribution reflected by lower VFT, WC and WHR. In contrast, boys had higher lean mass (FFM and BCM), reflecting sex-related differences in growth and body composition.

Interestingly, sex differences in body composition become evident as early as the first months of life, with males showing lower fat accumulation compared to females. Endogenous testosterone production in males between 1–4 months of age may explain this pattern and could have long-term effects on sex-specific body composition throughout life [38]. Another study found that before the age of twelve, body composition in boys and girls is relatively similar. However, after puberty, the differences between boys and girls become more pronounced, with boys generally increasing lean mass and girls accumulating more body fat [39]. Furthermore, estradiol concentrations were found to predict percentage body fat, indicating a role for estrogen in promoting fat storage [40]. During puberty, girls experience a more significant increase in adipose tissue compared to boys [41], likely influenced by hormonal changes [40]. Interestingly, in our cohort, fat mass levels did not differ significantly between girls and boys, suggesting differences in fat distribution may contribute more to CV risk than total fat content alone. This is further supported by the observation that girls in our cohort exhibited better CV risk profiles compared to boys. Our findings also align with a recent study, which found that adolescent boys with overweight or obesity are at a higher risk of abdominal fat accumulation and cardiovascular problems compared to girls. The study findings emphasized the importance of considering sex-specific risk factors when evaluating cardiometabolic risks and potential organ damage in overweight or obese youths [42].

Substantial evidence indicates that estrogen influences CV physiology and function in both health and disease, potentially acting as a cardioprotective factor [43]. Estrogen improves vascular health by promoting reendothelialization, reducing inflammation in endothelial cells, and regulating reactive oxygen species (ROS) production. It also inhibits vascular smooth muscle cell proliferation. Additionally, estrogen enhances the lipoprotein profile by increasing HDL-c and preventing LDL-c oxidation, which reduces the accumulation of oxidized LDL-c in the arterial wall [43,44,45]. Preclinical and clinical studies indicate that estrogenic agents influence early atherogenesis by interacting with the vascular immune system. Modified lipoproteins contribute to endothelial dysfunction, cholesterol accumulation, and driving macrophage-derived monocytes to produce inflammatory mediators and cytokines, all of which play a significant role in the development of atherosclerosis [46]. Interestingly, no significant differences in BP or PWV were observed between boys and girls in our cohort, suggesting that sex differences in vascular function may not yet be apparent. This could be due to insufficient time for age-related hormonal changes to significantly impair endothelial function and elevate CV risk. Further stratification by pubertal stage would be beneficial.

Nonetheless, our results indicate that boys exhibited a greater accumulation of cardiovascular risk factors compared to their female counterparts. Sex-differences in laboratory-based CV risk assessment were observed, with girls showing lower levels of liver enzymes (AST and ALT), uric acid, urea, cystatin C, TSH, T3, and hemoglobin. This is in concordance with studies showing that increased cardiovascular risk appears earlier in boys than in girls [47], with male adolescents with severe obesity displaying a more unfavorable set of metabolic and behavioral risk factors for cardiovascular disease compared to girls [48]. Furthermore, innovative cardiovascular risk markers such as the Triglyceride Glucose Index, Body Mass Index combined with the Triglyceride Glucose Index, Visceral Adiposity Index, Lipid Accumulation Product Index, Fatty Liver Index, and Hepatic Steatosis Index could help identify children at higher cardiovascular risk and provide additional insights into sex-related differences [49].

Obesity-induced low-grade chronic inflammation, along with the endocrine, paracrine, and metabolic effects of obesity, increases the long-term risk for several severe diseases [2]. Oxidative stress and inflammation are interconnected pathophysiological processes, with one often triggering the other. As a result, both processes are commonly present in various pathological conditions [50]. The overall state of inflammation and oxidative stress in our cohort showed a positive correlation with several anthropometric measurements of obesity, abdominal obesity indices such as WHtR, WC, UZ-measured VFT, and SFT at abdomen, and fat mass assessed with BIA. A clear negative correlation was observed between the inflammation/oxidative stress state and HDL levels. Interestingly, there was a positive relationship with PWV, indicating stiffening of arteries in relation to increased oxidative stress and inflammation.

MPO exhibited the most numerous correlations among the selected biomarkers. It was positively correlated with weight, pBMI, BMI, and hip circumference, as well as abdominal adiposity indices, including waist circumference, WHtR, visceral fat thickness, subcutaneous fat thickness at abdomen, and several body composition parameters, and negatively correlated with HDL levels. Interestingly, MPO levels were elevated in boys with obesity or overweight, but not in their female counterparts. These findings align with other studies that have also reported increased MPO levels in children and adolescents with obesity [51,52,53], but they cannot explain sex-related difference observed in our cohort. In our analysis, boys had more severe obesity than girls, which aligns with broader trends observed in high-income countries, where boys are more likely to develop such a condition [4]. This gender-related distribution of morbid obesity likely plays a key role in the observed differences in MPO levels. Based on several observed correlations and its evident elevation in children with obesity [27], we consider MPO a promising biomarker with potential for inclusion in the clinical workup in children and adolescents with obesity. Additionally, previous studies about MPO show promise not only for prevention and monitoring but also for therapeutic applications. Research has demonstrated that MPO deficiency or the use of MPO inhibitors can reduce inflammation and mitigate tissue injuries, indicating that MPO could serve as a valuable target for both prognostic and therapeutic strategies [54].

Furthermore, elevated MPO levels are linked to a higher risk of CV disease [51]. MPO is crucial in atherogenesis and contributes to endothelial damage in the early stages of the process [35,38,39]. It has specifically been implicated in CV risk through various mechanisms such as the release of reactive oxygen species, oxidation of LDL particles, transformation of HDL into dysfunctional particles, endothelial dysfunction, platelet aggregation, and the increased vulnerability of atheromatous plaques [51,55]. In our cohort, a significant correlation was observed between MPO and SBP, which is in concordance with those findings. There was also a trend toward positive correlation between MPO and PWV. As vascular stiffening may require a longer duration of obesity to become evident, a longitudinal study or stratification of participants by age could provide more comprehensive insights into these associations.

The other two biomarkers elevated in groups stratified by sex and increased pBMI were leukocytes and CRP. Both boys and girls with overweight or obesity had significantly higher CRP levels. Additionally, girls with excess pBMI showed increased leukocyte counts compared to their normal-weight counterparts. This aligns with other studies that highlight a positive relationship between leukocyte count and BMI, with a stronger association of obesity-related leukocytosis in females compared to males [56]. Furthermore, WBC showed a positive correlation with waist-to-height ratio (WHtR), while CRP was positively correlated with both WHtR, BMI, pBMI and WC. Their advantage lies in their accessibility, as they are not only used for research purposes but are also part of routine clinical practice. Our findings are consistent with studies suggesting that routine markers such as total leukocyte count and CRP may reliably reflect inflammation in children and adolescents with obesity [20,21,22,23].

On the other hand, the biomarkers MCP-1 and SOD-1 were not altered in our groups and showed no correlations with the selected adiposity indices, laboratory parameters, pulse wave velocity, or blood pressure. This may be due to the inclusion of children in the overweight range, who exhibit less pronounced cardiometabolic changes. This could also explain why adiponectin, the most abundant serum adipokine, which has previously shown an inverse correlation with abdominal obesity in children [57], was not statistically different among groups. However, ADPN showed a positive correlation with BMI and WC and negative correlation to HDL levels, indicating metabolic changes in obesity.

Additionally, we measured the levels of I-TAC, a novel biomarker linked to obesity and cardiovascular disease. Research suggests that certain members of the CXCL family, particularly CXCL10 and CXCL11, could serve as potential indicators of adipose tissue inflammation in obesity [58]. As CXC chemokines play a key role in recruiting leukocytes and macrophages and possess significant immunomodulatory properties, they may help predict the effectiveness of treatments for obesity and related conditions [59]. Increased I-TAC levels have been reported in adults with chronic obesity and related comorbidities [24,25], indicating that such elevation may be more advanced stages of obesity. Our finding aligns with this, as no significant difference in I-TAC levels was found between boys or girls with obesity and their normal-weight counterparts. The duration and severity of obesity in the pediatric population may not yet be sufficient to cause a significant rise in I-TAC levels. In contrast, studies in adults involved individuals with diverse metabolic health statuses and comorbidities, which may explain the elevated I-TAC levels found in those populations but not in our pediatric cohort. Future studies should incorporate larger, more diverse cohorts, ensuring greater representation of individuals with severe obesity, and consider the duration of obesity to better understand its cumulative impact on inflammation, oxidative stress, and health outcomes.

## 5. Conclusions

To conclude, this study highlights the associations between adverse body composition, inflammation, oxidative stress, and other cardiovascular risk factors in pediatric obesity. Notably, sex-related differences were observed in some inflammatory and oxidative stress biomarkers, body composition, fat distribution, and overall cardiovascular risk factor accumulation, with boys showing greater impairment compared to girls. These findings emphasize the importance of considering sex-related variations when assessing metabolic and vascular health in children. While the role of I-TAC/CXCL11 remains unclear and warrants further investigation, the combined use of biomarkers such as MPO, CRP, and WBC shows potential for identifying early inflammatory and oxidative stress changes, monitoring disease progression, and guiding future therapeutic approaches.

## Figures and Tables

**Figure 1 biomedicines-13-00058-f001:**
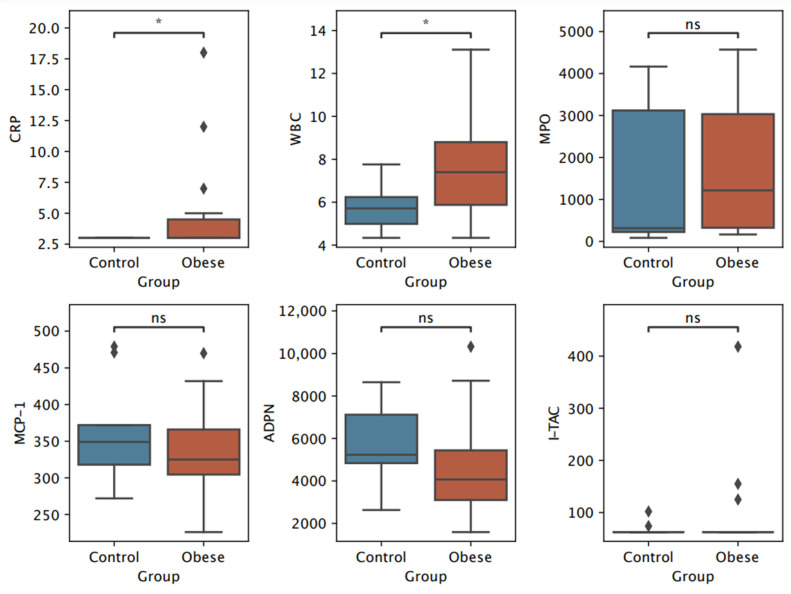
Boxplots of selected inflammation and oxidative stress biomarkers in girls stratified by BMI percentile (pBMI). *p*-value annotation legend: ns: Not significant (*p* > 0.05), *: Significant (0.01 < *p* ≤ 0.05). CRP—c-reactive protein, WBC—white blood cells/leukocytes, MPO—myeloperoxidase, MCP-1—monocyte chemoattractant protein-1, ADPN—adiponectin, I-TAC—interferon-inducible T-cell alpha chemoattractant.

**Figure 2 biomedicines-13-00058-f002:**
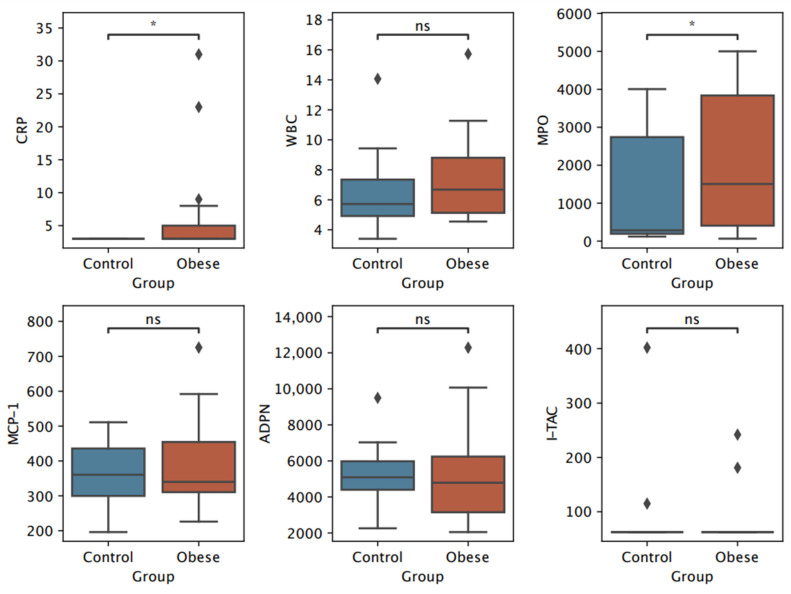
Boxplots of selected inflammation and oxidative stress biomarkers in boys stratified by BMI percentile (pBMI). *p*-value annotation legend: ns: Not significant (*p* > 0.05), *: Significant (0.01 < *p* ≤ 0.05). CRP—c-reactive protein, WBC—white blood cells/leukocytes, MPO—myeloperoxidase, MCP-1—monocyte chemoattractant protein-1, ADPN—adiponectin, I-TAC—interferon-inducible T-cell alpha chemoattractant.

**Figure 3 biomedicines-13-00058-f003:**
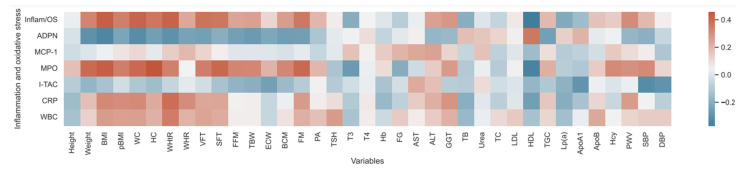
Heatmap for inflammation/oxidative stress parameters in correlation to adiposity indices, laboratory parameters, blood pressure and pulse wave velocity. Positive correlations are represented by shades of red, while negative correlations are indicated by shades of blue. Because of several correlations, BH correction was performed. Inflam/OS—overall inflammation and oxidative stress, ADPN—adiponectin, MCP-1—monocyte chemoattractant protein-1, MPO—myeloperoxidase, I-TAC—interferon-inducible T-cell alpha chemoattractant, CRP—c-reactive protein, WBC—white blood cells/leukocytes, BMI—body mass index, WC—waist circumference, HC—hip circumference, WHtR—waist-to-height ratio, WHR—waist-to-height ratio, VFT—visceral fat thickness, SFT—subcutaneous fat thickness, FFM—fat-free mass, TBW—total body water, ECW—extracellular water, BCM—body cell mass, FM—fat mass, PA—phase angle, TSH—thyroid-stimulating hormone, T3—triiodothyronine, T4—thyroxine, Hb—hemoglobin, FG—fasting glucose, AST—aspartate aminotransferase, ALT—alanine aminotransferase, GGT—gamma-glutamyl transferase, TB—total bilirubin, TC—total cholesterol, LDL—low-density lipoprotein, HDL—high-density lipoprotein, TGC—triglycerides, Lp (a)—ipoprotein (a), ApoA1—apolipoprotein A1, ApoB—apolipoprotein B, Hcy—homocysteine, PWV—pulse wave velocity, SBP—systolic blood pressure, DBP—diastolic blood pressure.

**Table 1 biomedicines-13-00058-t001:** Cohort characteristics with adiposity indices (girls vs. boys).

Parameters	Girls, Median (IQR)	Boys, Median (IQR)	*p*-Values
Age (years)	14 (3.25)	14 (4)	0.938
Weight (kg)	66.2 (24.7) *	81.8 (49.2) *	0.038
Height (cm)	165.5 (12.4) *	174.5 (18.3) *	0.002
Body mass index (kg/m^2^)	25.2 (9.5)	27.7 (11.5)	0.29
Body mass index percentile	95.5 (29.0)	97.2 (34.6)	0.135
Waist circumference (cm)	82.5 (22.5) *	92.0 (34.5) *	0.007
Hip circumference (cm)	94.0 (23.0)	103.5 (26.1)	0.168
Waist-height ratio	0.5 (0.1)	0.6 (0.2)	0.094
Waist-hip ratio	0.8 (0.1) *	0.9 (0.1) *	<0.001
UZ-measured visceral fat thickness (mm)	46.4 (15.3) *	56.3 (28.8) *	0.002
UZ-measured subcutaneous fat thickness (mm)	23.5 (22.4)	31.0 (36.2)	0.523
Fat-free mass (kg)	45.9 (15.4) *	56.0 (26.0) *	0.003
Total body water (L)	34.8 (11.9) *	42.0 (16.2) *	<0.001
Extracellular water (L)	14.8 (4.2) *	18.0 (7.4) *	0.003
Body cell mass (kg)	24.9 (9.7) *	30.8 (17.4) *	0.003
Fat mass FM (kg)	20.8 (17.9)	25.0 (24.4)	0.36
Phase angle (°)	6.1 (0.6) *	6.6 (0.9) *	0.018

*—statistically significant difference.

**Table 2 biomedicines-13-00058-t002:** Blood pressure, pulse wave velocity, and laboratory tests for cardiovascular risk evaluation (girls vs. boys).

Parameters	Girls, Median (IQR), N = 36 (45%)	Boys, Median (IQR), N = 44 (55%)	*p*-Values
Systolic blood pressure (mmHg)	124.5 (22.5)	122.0 (21.5)	0.877
Diastolic blood pressure (mmHg)	76.5 (15.5)	75.0 (14.5)	0.276
Pulse wave velocity (m/s)	6.2 (1.4)	6.2 (1.6)	0.82
Thyroid-stimulating hormone (mU/L)	2.0 (1.0) *	2.4 (1.3) *	0.011
Triiodothyronine (pmol/L)	5.6 (0.6) *	6.1 (1.2) *	0.001
Thyroxine (pmol/L)	15.7 (2.9)	15.4 (2.3)	0.77
Fasting glucose (mmol/L)	4.0 (0.5)	4.7 (0.5)	0.168
Aspartate aminotransferase (µkat/L)	0.3 (0.1) *	0.4 (0.2) *	0.002
Alanine aminotransferase (µkat/L)	0.4 (0.2) *	0.5 (0.3) *	<0.001
Gamma-glutamyl transferase (µkat/L)	0.4 (0.1)	0.4 (0.2)	0.062
Total bilirubin (µmol/L)	9.5 (7.0)	9.5 (7.0)	0.538
Urea (mmol/L)	3.8 (1.1) *	4.7 (1.1) *	<0.001
Creatinine (mmol/L)	55.5 (13.5)	57.5 (24.3)	0.096
Urate (µmol/L)	277.5 (77.0) *	330.0 (124.0) *	0.04
Cystatin C (mg/L)	0.8 (0.2) *	1.0 (0.2) *	<0.001
Total cholesterol (mmol/L)	3.6 (1.1)	4.0 (0.7)	0.229
Low-density lipoprotein cholesterol (mmol/L)	2.3 (0.7)	2.4 (0.7)	0.325
High-density lipoprotein cholesterol (mmol/L)	1.3 (0.5)	1.3 (0.4)	0.52
Triglycerides (mmol/L)	0.8 (0.7)	0.9 (0.8)	0.277
Lipoprotein (a) (mg/dL)	101.0 (275.5)	78.5 (206.0)	0.175
Apolipoprotein A1 (g/L)	1.4 (0.3)	1.5 (0.2)	0.681
Apolipoprotein B (g/L)	0.7 (0.2)	0.7 (0.2)	0.216
Homocysteine (µmol/L)	8.1 (2.8)	8.6 (3.2)	0.095
Hemoglobin (g/L)	134.0 (9.3) *	145.5 (21.0) *	<0.001

*—statistically significant difference.

**Table 3 biomedicines-13-00058-t003:** Selected inflammation and oxidative stress biomarkers (girls vs. boys).

Parameters	Girls, Median (IQR), N = 36 (45%)	Boys, Median (IQR), N = 44 (55%)	*p*-Values
Leukocytes (×10^9^/L)	6.4 (3.0)	6.2 (3.0)	0.854
C-reactive protein (mg/L)	3 (0)	3 (0)	0.839
Myeloperoxidase (ng/mL)	580.0 (2835.0)	937.5 (2873.8)	0.866
Adiponectin (ng/mL)	4900.0 (2725.0)	4965.0 (2855.0)	0.798
Monocyte chemoattractant protein-1 (pg/mL)	335.5 (61.8)	345.0 (138.0)	0.156
Interferon-inducible T-cell alpha chemoattractant (pg/mL)	62.5 (0)	62.5 (0)	0.555

**Table 4 biomedicines-13-00058-t004:** Selected inflammation and oxidative stress biomarkers based on sex and BMI percentile (pBMI).

Parameters	Girls with Normal Weight, Median (IQR)	Girls with OverWeight/Obesity, Median (IQR)	*p*-Values	Boys with Normal Weight, Median (IQR)	Boys with OverWeight/Obesity, Median (IQR)	*p*-Values
Leukocytes (×10^9^/L)	5.7 (1.3) *	7.5 (2.2) *	0.026	5.7 (2.4)	6.7 (3.7)	0.177
C-reactive protein (mg/L)	3 (0) *	3 (0) *	0.039	3 (0) *	3 (2) *	0.03
Myeloperoxidase (ng/mL)	315.0 (2895.0)	1215 (2707.5)	0.157	285.0 (2545.0) *	1505 (3430.0) *	0.017
Adiponectin (ng/mL)	5230.0 (2280.0)	4070.0 (2340.0)	0.075	5090.0 (1580.0)	4790.0 (3090.0)	0.588
Monocyte chemoattractant protein-1 (pg/mL)	349.0 (54.0)	325.0 (61.5)	0.459	360.5 (136.0)	340.0 (144.0)	0.831
Interferon-inducible T-cell alpha chemoattractant (pg/mL)	62.5 (0)	62.5 (0)	1	62.5 (0)	62.5 (0)	0.646

*—statistically significant difference.

## Data Availability

The original contributions presented in this study are included in the article. Further inquiries can be directed to the corresponding author.

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
