# Peer review of "Systemic Inflammation and Oxidative Stress in Childhood Obesity: Sex Differences in Adiposity Indices and Cardiovascular Risk"

_biomedicines, 2024, doi:10.3390/biomedicines13010058_

Round 1

Reviewer 1 Report

Comments and Suggestions for Authors

The authors conducted an interesting study aimed at assessing cardiovascular risk in childhood obesity by examining sex differences in adiposity indices, cardiometabolic profiles, inflammation, oxidative stress biomarkers, and the potential role of a novel molecular biomarker. The study analyzed eighty children, emphasizing that adiposity indices are crucial for evaluating cardiovascular risk in children and adolescents.   

This manuscript presents significant findings, especially considering recent global reports indicating a sharp increase in childhood obesity and cardiovascular risk among pediatric patients. Furthermore, it is well-known that cardiovascular risk factors, such as being overweight, exert an even greater harmful effect when acquired during childhood.  

Below are my comments:  

- Line 41: It would be helpful to include the diagnostic criteria for metabolic syndrome in both adult and pediatric populations, as the latter has specific and well-defined criteria. This is particularly relevant since metabolic syndrome is often associated with overweight and obesity.  

- Line 62: Specify the anatomical sites at higher risk for atherosclerosis in the pediatric population, as these differ from the commonly affected sites in adults.  

- Statistical methods and sample size calculation: These are described in detail.  

- Ethical approval: The ethical approval process is appropriately documented.  

- Line 166: Add reference parameters for interpreting blood pressure values.  

- Lines 185, 193, and 201Include p-values in the tables. The term "statistically significant difference" is misleading if the level of significance cannot be determined. This observation applies to all other result tables as well.  

- Line 283: There are some relevant studies available that have analyzed the same indices as the authors, alongside other innovative cardiovascular risk markers commonly applied in adult populations and currently being validated in pediatric populations (e.g., 10.3390/diseases12060119). These could enhance the discussion.  

- Line 318: From the authors' perspective, can this statement also be considered valid during adolescence?  

- Line 328: Were blood pressure parameters between males and females assessed using the same percentiles?  

The conclusions align with the objectives of the manuscript.  

Minor improvements in English language usage throughout the manuscript are recommended.  

Reviewer 2 Report

Comments and Suggestions for Authors

Recommendations:

1. Really high iThenticate-need to reduce it.

2. I doubt that the difference in MPO are so dramatic between obese and non obese boys.

3. However, could you elaborate on the role of Neutrophils and platelets in systemic inflammation in these patients? see this: https://doi.org/10.3390/cimb46080496

4. Statistics methodology, remove it from results and put it in the methods section.

5. A more accurate analysis could have been a Kruskal-Wallis test evaluating the values of inflammatory markers between all BMI classes (i.e. normal, overweight, type 1 obese, type 2 ..) overall, and gender distributed.

6.  There are some cases selection biases, a person overweight is very different from an type 2 obsese or type 3, how do you confront this bias? 

Round 2

Reviewer 1 Report

Comments and Suggestions for Authors

The authors have adequately addressed all my comments. The manuscript has significantly improved compared to the previous version, and I believe it can be accepted in its current form, pending the final decision of the Editor.